# The Artemiside-Artemisox-Artemisone-M1 Tetrad: Efficacies against Blood Stage *P. falciparum* Parasites, DMPK Properties, and the Case for Artemiside

**DOI:** 10.3390/pharmaceutics13122066

**Published:** 2021-12-03

**Authors:** Liezl Gibhard, Dina Coertzen, Janette Reader, Mariëtte E. van der Watt, Lyn-Marie Birkholtz, Ho Ning Wong, Kevin T. Batty, Richard K. Haynes, Lubbe Wiesner

**Affiliations:** 1H3D, Department of Chemistry, University of Cape Town, Observatory, Cape Town 7925, South Africa; liezl.gibhard@uct.ac.za; 2Malaria Parasite Molecular Laboratory, Department of Biochemistry, Genetics and Microbiology, Institute for Sustainable Malaria Control, University of Pretoria, Pretoria 0028, South Africa; dina.coertzen@up.ac.za (D.C.); janette.reader@up.ac.za (J.R.); mariette.vanderwatt@up.ac.za (M.E.v.d.W.); lynmarie.birkholtz@up.ac.za (L.-M.B.); 3Centre of Excellence for Pharmaceutical Sciences, Faculty of Health Sciences, North-West University, Potchefstroom 2520, South Africa; 22966390@nwu.ac.za; 4Curtin Medical School, Curtin University, Bentley 6102, Australia; Kevin.Batty@curtin.edu.au; 5Division of Clinical Pharmacology, Department of Medicine, University of Cape Town, Observatory, Cape Town 7925, South Africa

**Keywords:** antimalarial drugs, artemisinins, ACTs, resistance, amino-artemisinins, pharmacokinetics, metabolism, C_max_, drug efficacy

## Abstract

Because of the need to replace the current clinical artemisinins in artemisinin combination therapies, we are evaluating fitness of amino-artemisinins for this purpose. These include the thiomorpholine derivative artemiside obtained in one scalable synthetic step from dihydroartemisinin (DHA) and the derived sulfone artemisone. We have recently shown that artemiside undergoes facile metabolism via the sulfoxide artemisox into artemisone and thence into the unsaturated metabolite M1; DHA is not a metabolite. Artemisox and M1 are now found to be approximately equipotent with artemiside and artemisone in vitro against asexual *P. falciparum* (*Pf*) blood stage parasites (IC_50_ 1.5–2.6 nM). Against *Pf* NF54 blood stage gametocytes, artemisox is potently active (IC_50_ 18.9 nM early-stage, 2.7 nM late-stage), although against the late-stage gametocytes, activity is expressed, like other amino-artemisinins, at a prolonged incubation time of 72 h. Comparative drug metabolism and pharmacokinetic (DMPK) properties were assessed via *po* and *iv* administration of artemiside, artemisox, and artemisone in a murine model. Following oral administration, the composite C_max_ value of artemiside plus its metabolites artemisox and artemisone formed in vivo is some 2.6-fold higher than that attained following administration of artemisone alone. Given that efficacy of short half-life rapidly-acting antimalarial drugs such as the artemisinins is associated with C_max_, it is apparent that artemiside will be more active than artemisone in vivo, due to additive effects of the metabolites. As is evident from earlier data, artemiside indeed possesses appreciably greater efficacy in vivo against murine malaria. Overall, the higher exposure levels of active drug following administration of artemiside coupled with its synthetic accessibility indicate it is much the preferred drug for incorporation into rational new artemisinin combination therapies.

## 1. Introduction

Given the problems both of enhanced tolerance of the malaria parasite *Plasmodium falciparum* (*Pf*) to the current clinical artemisinins (Figure 1) and formal resistance to the antimalarial partner drug in artemisinin-based combination therapies (ACTs) [1,2,3,4,5], rational new triple artemisinin-based combination therapies (TACTs) based on discrete consideration of mechanism of action of the components are urgently required [6,7]. We thereby focus on newer artemisinin derivatives that are pan-reactive against blood stage parasites, a redox active second drug such as methylene blue, phenoxazine or naphthoquinone which displays synergism with the artemisinins [6,7], and a third drug type such as a quinolone that has a distinct target [8]. The non-artemisinin component must be active against ring-stage dormant parasites associated with the artemisinin-tolerant phenotypes. Of the current clinical artemisinin derivatives that were all originally prepared by Chinese scientists working under the remarkable Project 523 [9,10,11,12,13], artemether **3** and artesunate **4** undergo metabolism or hydrolysis respectively to DHA **2** [14,15,16]. The process is so facile for artesunate that this acts as a prodrug for DHA [17,18,19,20]. DHA is labile under physiological conditions [21,22,23] or in water at pH 7.2 wherein decomposition prevents measurement of solubility [24,25]. This intrinsic chemical instability coupled with its implicit involvement in artemisinin resistance [26,27] and its neurotoxicity [28,29] are not reconcilable with its continued use as a frontline antimalarial drug [30]. Even though it now appears that within certain of the current ACTs, the partner drug acts to enhance activity of the artemisinin against artemisinin-tolerant strains, e.g., artemether-lumefantrine [31] or artesunate-pyronaridine [32], the latter in a clinical situation, artemisinin derivatives that do not decompose to DHA are required.

Amino-artemisinins bear an amino-alkyl or other amino group attached via the nitrogen atom to C-10 [24,33,34] wherein this group enhances the antimalarial mechanism of action [35] and prevents metabolism to DHA [24,25]. One such is artemiside **5** that is obtained in one scalable step from DHA [24,36] and, depending upon the oxidizing agent used, is converted either into the sulfoxide artemisox **6** or the sulfone artemisone **7** (Figure 1) [24]. The last was selected as a development candidate on the basis of its efficacy profile, relatively benign toxicity [24] including lack of neurotoxicity [37,38], and in notable contrast to current clinical artemisinins [14,17,19], lack of induction of its own metabolism [24,39]. Artemisone **7** is metabolized principally by desaturation of the thiomorpholine-*S*,*S*-dioxide to metabolites M1 **8** and by hydroxylation of artemisone at C5 and C7 to M2 and M3 respectively (Figure 1b) [24,25]. According to ex vivo bioassay in a primate model using healthy *Aotus* monkeys, the metabolites are biologically active and contribute substantially to overall antimalarial activity. Activity is observable up to 7 h after administration of a single dose of artemisone **7** (10 mg/kg); in comparison, equivalent activity of a single dose of artesunate **4** (10 mg/kg) is observable up to 1 h post dose where the active metabolite must be DHA **2 [24]**. In a clinical Phase I trial of artemisone **7**, ex vivo bioassay of subject plasma samples displayed antimalarial activity 2.3-fold higher at *T*_max_ than that corresponding to the actual plasma concentration of artemisone as determined by liquid chromatography-tandem mass spectrometry (LC-MS/MS) [39]. The enhanced antimalarial activity was due to the active metabolites M1, M2, and M3 and minor amounts of other metabolites. Notably, M1 **8** possesses over a 15-fold greater human microsomal half-life than does artemisone [25] and is active against multidrug resistant *Pf* in vitro [40].

We have recently shown that artemiside **5** is metabolized to artemisone **7** and thence to M1 **8**; no attempt was made in this study to track formation of the other possible metabolites (*cf.*
Figure 1) [25]. However, the biotransformation in resembling that of other drugs with sulfide linkages must proceed via the sulfoxide artemisox **6** [41,42,43]. We now report the results of the evaluation of the efficacies of artemisox and M1 against *Pf* blood stages with screening protocols previously used for artemiside and artemisone [6,7], cytotoxicities against mammalian HepG2 cells, and comparative drug metabolism-pharmacokinetic (DMPK) studies on the drug triad of artemiside **5**, artemisox **6,** and artemisone **7**. The data provide a clear indication of which drug in the foregoing triad is most suitable for using as the artemisinin component in new TACTs.

## 2. Results

### 2.1. Artemisox Efficacy

#### 2.1.1. Asexual Blood Stage Parasites

The inhibitory concentrations of artemisox, metabolite M1, and chloroquine and DHA, artemether, and artesunate were determined using the SYBR Green I based assay on asexual blood stages of *Pf* NF54 (drug sensitive) and K1 and W2 (drug resistant) strains. Table 1 presents the activities and the resistance index (RI) for each drug resistant strain. Dose response curves are in Appendix A. Table 1 also includes the historical values for artemiside and artemisone obtained using the same assay [6,7]. Overall, the IC_50_ values of ~2 nM for artemisox **6** and M1 **8** closely cohere with those displayed by each of artemiside **5** and artemisone **7**. However, multidrug resistant parasites were slightly more sensitive to artemisox, as indicated by the RI ratios of <1. Activities of artemisone **7** and M1 **8** obtained using a different assay method against other multidrug resistant strains, including atovaquone resistant strains (Appendix A), are comparable [40]. In line with previous reports involving assessment of cytotoxicities of antimalarial drugs against the mammalian liver HepG2 cell line [44], activities of the amino-artemisinins in Table 1 are some five orders of magnitude less than against the malarial parasites.

#### 2.1.2. Blood Stage Gametocytes

For screening artemisox and M1 against early- (EG) and late-stage (LG) gametocytes, the luciferase assay with transgenic NF54-*pfs*16-GFP-Luc and NF54-Mal8p1.16-GFP-Luc parasite lines was used [6,7,45]. With this assay, stage-specificity of gametocytocidal action was established (Table 2). Dose-response curves are in Appendix A. Artemiside, artemisox, artemisone, and M1 all possessed activities against early-stage gametocytes superior to those of DHA, artemether, and artesunate [6]. As was observed previously for artemisone and artemiside, dose responses for artemisox and M1 could not be obtained at 48 h against late-stage gametocytes [6,7]. However, at a 72 h incubation period, artemisox displayed activity against late-stage gametocytes similar to that of artemiside and superior to that of artemisone [6]. Despite showing comparable activity on early-stage gametocytes and a >90% inhibition against late-stages at 48 h, dose response for M1 could not be obtained at 72 h. The metabolic precursor of M1, artemisone, showed a significant (*p* < 0.05) >20-fold decrease in activity against late-stage gametocytes (Table 2).

### 2.2. Metabolism and Pharmacokinetics of Artemiside, Artemisox and Artemisone

The plasma concentration-time profiles of the parent compounds and metabolites after administration *iv* and *po* to male C57/BL6 mice are presented in Figure 2, Figure 3 and Figure 4. Artemiside was quantifiable up to 24 h in all mice dosed *iv* and up to 7 h *po* (lower limit of quantitation LLOQ = 1.6 ng/mL or 0.004 μM) (Figure 2). Administration of artemisox also resulted in its rapid conversion into artemisone, although it was detectable up to 7 h in the *po* group and up to 5 h in the *iv* group (Figure 3). For both artemiside and artemisox, the metabolite artemisone was quantifiable up to 3 h in both the *po* and *iv* groups (Figure 2 and Figure 3). Thereby it is confirmed that artemiside is rapidly converted via artemisox into artemisone. Artemisone itself was quantifiable up to three hours post dose (Figure 4). The mean circulating concentrations for the compounds and their metabolites are presented in Appendix A. The metabolite M1 was also detected upon administration of artemiside, artemisox, and artemisone, but it was not quantified here; chromatograms indicating the presence of M1 are illustrated in Appendix A. In addition, the plasma concentration-time curves for artemiside and its metabolites artemisone and M1 following *iv* and *po* dosing in the same murine model are presented and discussed in detail elsewhere (*cf.* Figure 4 in reference [25]) [25].

### 2.3. Pharmacokinetic Parameters

For the *iv* dose, artemiside displayed a clearance from plasma (3.5 L/h/kg) that is rated as moderate [46]. Artemisox and artemisone possessed higher systemic clearance (6.9 and 5.3 L/h/kg) and all three drugs were moderately distributed into tissues (3–4 L/kg). All compounds, as previously noted for artemiside and artemisone [25], had short half-lives and mean residence times and were rapidly absorbed following oral administration. Whilst oral exposure was low for artemiside and artemisone, artemisox had a moderate oral exposure. Data are summarized in Table 3 and Table 4.

For analysis of this data, we consider pharmacokinetic parameters of the parent compounds alone and in conjunction with the metabolites. The standout features are as follows. Based on AUC_0-last_ values, oral administration of artemiside **5** results in 2.5 times higher exposure for artemisone **7**, compared to the situation when artemisone **7** is administered alone (Figure 2 and Figure 4 and Table 4). Following oral dosing of artemisox **6** (Figure 3), the oral exposure of artemisone **7** is 4.3 times higher than dosing artemisone **7** alone (Table 4).

## 3. Discussion

The rapid metabolism via oxidative conversion of the sulfide in the thiomorpholine ring of artemiside to the thiomorpholine-*S*,*S*-dioxide of artemisone proceeds via the sulfoxide artemisox. This compound possesses antimalarial efficacies in vitro against *Pf* asexual and sexual blood stages directly comparable with those of artemiside and artemisone (Table 1 and Table 2).

### 3.1. Metabolism of Artemiside ***5***

The oxidation of the sulfide group in endogenous compounds or xenobiotics to the sulfoxide and then sulfone is well established. The best-known example of the former is the amino acid methionine within protein substrates, wherein it is rapidly oxidized non-enzymatically to methionine sulfoxide by reactive oxygen species (ROS). Methionine sulfoxide is readily reduced by methionine sulfoxide reductase [47,48]. However, further oxidation of methionine sulfoxide provides methionine sulfone that cannot be so reduced [49]. Analogous metabolic transformations are well established for drugs containing sulfide linkages, including cyclic sulfides closely related to those described here [50]. The anthelmintic albendazole incorporating a propylthio group attached to a benzimidazole undergoes rapid metabolism involving oxidation of the sulfide to the bioactive albendazole sulfoxide and then to albendazole sulfone [41]. Likewise, the anti-trypanosomial drug fexinidazole carrying a (4-methylthio)phenoxyl group attached to a nitroimidazole is metabolized via the corresponding sulfoxide to the sulfone, both more polar metabolites that in contrast to fexinidazole elicit blood concentrations above their effective therapeutic doses against the non-apicomplexan parasite *Leishmania donovani* [42,43]. However, artemiside is not a prodrug; as noted elsewhere [6,7] and as recorded here (Table 1 and Table 2), it possesses intrinsic activity against all *Pf* blood stages and is active against the apicomplexan parasite *Toxoplasma gondii* as summarized below.

The conversion of artemiside via artemisox to artemisone may proceed non-enzymatically, such as through oxidation by ROS or enzymatically by CYP enzymes such as in liver microsomes (Figure 5). The conversion of artemisone into the metabolite M1 and the hydroxylated artemisone metabolites M2 and M3 has been thoroughly studied using radiolabelled artemisone and principally involves CYP3A4 [24,39]. As noted above, these metabolites substantially add to overall efficacy of artemisone as established in ex vivo bioassays of monkey plasma [24] and of human plasma samples taken from subjects in a Phase I clinical study [39]. We show here that M1 is active in vitro against asexual blood stages of sensitive and multidrug resistant *Pf* (Table 1); activity against other multidrug resistant strains using a different assay method has been recorded previously [40] and is given in Appendix A. Further, solubility of M1 in aqueous solutions at different pH values is substantially greater than that of artemiside, and in contrast to artemiside and artemisone, possesses at least a 15-fold greater human microsomal half-life (Figure 5) [25]. Overall, the metabolism results in the conversion of a lipophilic drug with low oral bioavailability (artemiside *F%* 1; cf. artemether *F%* 2) into polar, more water-soluble derivatives, with greatly enhanced oral bioavailability, e.g., artemisone (*F%* 34) [25].

### 3.2. Pharmacokinetics

Whereas long half-life antimalarial drugs such as chloroquine may be more efficiently administered in multiple dose regimens over designated periods of time so as to keep drug levels in the blood above a defined minimum inhibitory concentration, for short half-life rapidly-acting drugs such as artemisinin and artesunate, it is established that administration as a single bolus dose is more efficient in ablating parasitaemia than administration via constant infusion; thus the driver of efficacy for artemisinins relates directly to C_max_ [51]. In the present study, oral administration of artemiside delivered a C_max_ (1.8 µM) for artemisone which is directly comparable to C_max_ (1.7 µM) for orally-administered artemisone (Table 4). However, when the combined total C_max_ values of orally-administered artemiside together with those of its metabolites artemisox and artemisone formed in situ are taken into consideration, the C_max_ of 4.4 µM is 2.6-fold higher compared to that of orally-administered artemisone. By comparison, the combined total C_max_ of orally administered artemisox and its metabolite artemisone (8.3 µM) is 4.9-fold higher than orally administered artemisone (Table 4). Therefore, in addition to the higher total exposure levels observed following administration of artemiside or artemisox *po*, the longer half-life (1.2 h vs. 0.4 h) and higher exposure provided by artemiside *iv* (Table 3) indicate that artemiside will likely be superior to artemisone as a clinical drug. Notably, as evident in Figure 2a, artemiside is rapidly converted to artemisone (formation half-life approximately 10 min) and its non-renal clearance also may be advantageous.

Direct comparison of our pharmacokinetic data with those from other animal studies of the clinical artemisinin derivatives such as dihydroartemisinin, artesunate, and artemether requires cautious interpretation due to use of different animal species and differences in the respective study designs [52,53,54,55]. In the case of artemether, we previously presented data indicating its rapid metabolism to DHA using oral administration at the same dose with the same murine model as described here (see Table 3, ref. [25]). Summation of C_max_ values for artemiside and its metabolites artemisone and M1 is approximately 1.67 times greater than that for artemether (C_max_ 0.6 µM, ref. [25]). However, no measurement of C_max_ for artemisox formed in situ was carried out in that study [25]. From the current study, summation of C_max_ values for artemiside and its metabolites artemisox and artemisone at 4.43 µM (Table 4) is some 7.4 times greater than C_max_ for artemether obtained from the earlier study. Thus, overall, the substantially higher exposure of artemiside and its active metabolites coupled also with their potency (Table 1) should result in enhanced activity with respect to artemether itself. We do not have comparable data for artesunate, but as noted above for artemisone itself, ex vivo bioassay in a primate model indicates that activity is observable up to 7 h after administration of a single dose of artemisone (10 mg/kg). Equivalent activity of a single dose of artesunate (10 mg/kg) is observable up to 1 h post dose where the active metabolite must be DHA [24]. Thus, it is clear that use of artemiside in this model will provide better results. This facile metabolism of artemiside to the non-neurotoxic artemisone in situ, coupled with consideration of the efficacies of artemiside itself and of the intermediate sulfoxide and of artemisone and M1 formed in situ, focusses attention on using artemiside as a clinical surrogate for artemisone. Thus, we must briefly review efficacy data for artemiside and artemisone in the context of total exposure of active drug according to the pharmacokinetic properties established here.

### 3.3. Comparison of Efficacies of Artemiside **5** and Artemisone **7**

Artemiside and artemisone are equipotent against asexual blood stages of sensitive and multidrug resistant *Pf* strains in vitro, with the metabolite M1 being slightly less active (Table 1) [6,7]. Activities against asexual blood stages of artemisinin-resistant *Pf* clones carrying the *Pf*KI3 C580Y mutation range from IC_50_ 0.27 to 2.43 nM [7], thus retaining efficacy levels characteristic of artemisinin-sensitive parasites (Appendix A); this is also recorded by others for artemisone [56,57]. Artemiside and artemisone are highly active against early-blood stage *Pf* NF54 gametocytes [6,7] as assessed using the luciferase-based assay originally used to assess the transmission blocking capabilities of methylene blue (Table 2) [45,58]. However, expression of activities of the amino-artemisinins against late-stage gametocytes is markedly dependent upon incubation times. Dose-responses for these compounds could not be obtained at 48 h, in contrast to the case with the early-stage assays, but rather at 72 h. In general, the efficacy of artemisinin derivatives is dependent upon incubation times and gametocyte stage specificity, and potency is only indicated at 48 h using the luciferase [6,59] and other assays [60,61]) when stage IV gametocytes are present. On the other hand, for mature stage V gametocytes, potency is not observed at 24–48 h [6,62], but requires a prolonged incubation period of 72–96 h [6,59,60]. Activities against the liver sporozoite stage of *P. berghei* parasites are IC_50_ 81.3 nM for artemiside and 28.3 nM for artemisone; in comparison the activity of artemether **3** is >10^4^ (Appendix A) [7].

The in vivo activity against murine malaria bears out post facto the pharmacokinetic prediction that artemiside, based on exposure as described above, is some threefold superior to artemisone. Artemiside administered subcutaneously (*sc*) or orally (*po*) to mice infected with *P. berghei* displays activities (ED_90_, mg/kg) according to the Peters four-day test of 0.51 and 1.9 mg/kg respectively; artemisone (ED_90_ *sc* 1.5, *po* 3.1 mg/kg) is less active. In comparison artesunate displays activities of 7.2 and 7.1 mg/kg (Appendix A) [24]. Artemiside (ED_90_ *sc* 0.61, *po* 2.0 mg/kg) is also more active than artemisone (ED_90_ *sc* 3.9, *po* 5.0 mg/kg) against chloroquine-resistant *P. yoelii* (Appendix A). The activity difference in vivo is also demonstrated in a model of cerebral malaria (CM) in mice infected with *P. berghei* ANKA strain [63,64]. Infected mice were treated on day 2 post infection by intraperitoneal (*ip*) injection of each of DHA (20 mg/kg/d), artesunate (10 mg/kg/d), artemiside (3 mg/kg/d), and artemisone (5 mg/kg/d) for 4 days (days 2–5 post-infection) [65]. Only artemiside induced complete cure, and there was no recrudescence after 25 days. In order to induce complete cure for artemisone, that is, no recrudescence on follow-up, twice daily doses of 5 mg/kg artemisone administered from day 3 post infection had to be used. In order to evaluate efficacies of drug combinations, the effects of each of artemisone and artemiside with piperaquine on the *P. berghei* ANKA infections in mice were examined [66]. When infected mice were treated *ip* with artemisone at 0.66 mg/kg and piperaquine at 10 mg/kg, reduction of the initial peak parasitemia took place, but recrudescence with eventual mortality was observed. In contrast, administration *ip* of the combination of artemiside and piperaquine at the foregoing dose levels resulted in survival of all mice, with no recrudescence taking place.

The superior activity of artemiside in vivo is also evident during treatment of mice infected with the virulent rapidly growing type I RH strain of the apicomplexan parasite *Toxoplasma gondii* (*Tg*) [67]. As for *Pf*, the compounds are equipotent in vitro, although activities are at least an order of magnitude inferior to activities against *Pf*. Thus, EC_50_ values of 0.108 µM and 0.120 µM respectively are recorded for each of artemiside and artemisone against *Tg* in vitro (*cf.* artesunate 0.213 µM). For assay of the acute phase of toxoplasmosis in vivo, mice infected by *ip* injection of type II tachyzoites (PRU-Luc-GFP) were treated *sc* with artemisinin **1**, artemiside, or artemisone, each 10 mg/kg/d, for 8 days. Whilst all mice treated with artemisinin succumbed to infection, 60% of the artemiside-treated mice and slightly over 50% of the artemisone-treated mice survived the infection. To evaluate effects on the reactivation stage of the disease, inbred gamma interferon IFN-γ^−/−^ female mice susceptible to both the acute and reactivation phases were used. The protocol involved challenge with parasites followed by treatment with sulfadiazine to suppress acute infection and allow development of tissue cysts. After three weeks, sulfadiazine treatment was stopped to allow for reactivation wherein conversion of bradyzoites to tachyzoites occurs. Following cessation of treatment with sulfadiazine, treatment with each of artemiside and artemisone (10 mg/kg/d) for 8 days prolonged survival in 80% and 60% respectively of the mice. Because IFN-γ^−/−^ mice cannot control proliferation of tachyzoites, the control mice died within 10 days. Thus, although the two compounds controlled the initial phases of reactivated infection, they could not achieve complete cure. When treatment with the compounds was discontinued, all mice succumbed by day 25. Thus, both compounds are not able to eradicate the chronic infection and likely do not act directly on bradyzoites. The outcome resembles that of treatment with atovaquone; overall, the efficacies of artemiside and artemisone compare favorably with that of atovaquone. Thereby, the potential emerges for use of artemiside in combination with pyrimethamine or clindamycin for treatment of patients that cannot tolerate sulfadiazene, normally used together with pyrimethamine for treatment of toxoplasmosis [66].

### 3.4. Comparative Toxicities of Artemiside **5** and Artemisone **7**

The important issue associated with the in vivo studies is the toxicity, including neurotoxicity, of artemiside and artemisone relative to the current clinical artemisinins. Artemiside is more lipophilic (Log *P* 4.97, CLog *P* 3.98) than is artemisone (log *P* 2.49, CLog *P* 2.08; Figure 5). In accord with the precept that higher intrinsic lipophilicity of an artemisinin derivative imprints greater neurotoxicity [68], artemiside, in contrast to artemisone, does possess neurotoxic potential. A quantitative assay customized to evaluate neurotoxicity of artemisinins involves use of fetal rat brain stem cells cultured to generate permanent neuronal networks for 8 days. Compounds with known neurotoxins as comparators are applied on day 9 and effects are examined over the following 7 days [28,37,69]. In this assay, artemisone displays negligible effects on viability, ATP levels, or on neurofilament outgrowth (Appendix A) [24]. However, although artemiside displays a non-observable effect concentration (NOEC) of 1 µg/mL on viability and ATP levels, these are at least an order of magnitude greater, that is, less toxic than those displayed by DHA and artesunate (each 0.1 µg/mL). Neurofilament outgrowth is particularly affected by DHA (NOEC < 0.001 µg/mL, IC_50_ 0.01 µg/mL). These values reflect the potent neurotoxicity of DHA and indeed approximate the level of antimalarial efficacies of DHA in vitro against *Pf* (*cf*. Table 1 and Appendix A). Effects exerted by artemiside, particularly as reflected in the IC_50_ value, are significantly less (Appendix A) [24]. For evaluation of neurotoxicity in vivo*,* male rats were treated with artemiside or artemisone at 10 mg/kg/d for 14 days by gavage in sesame oil, a vehicle that exacerbates neurotoxic effects [70]. However, for both drugs, no effects were observed. When administered at 50 mg/kg/d for 14 days, artemiside elicited neurotoxic sequelae including body weight loss, reduced motility, uncoordinated gait, and piloerection; in contrast at this dose level, artemisone induced no effects, and weight gains were as for the controls [24,37]. It is this notable lack of toxicity that was a key determinant in the original selection of artemisone as a clinical development candidate [24]. Finally, results from in vitro assays involving other cell lines including Chinese hamster ovary (CHO) cells indicate that artemiside and artemisone (EC_50_ > 241 µM) are an order of magnitude less cytotoxic than DHA (EC_50_ 25.2 µM) [6,7].

## 4. Materials and Methods

Artemiside, artemisone, and metabolite M1 are from batches used for efficacy assays [6,7] and DMPK analysis [25]. Crystalline artemisox was prepared by oxidation of artemiside with *m*-chloroperbenzoic acid in diethyl ether and purified by recrystallization from isopropanol [24]. Reference compounds and compounds for screening were ≥95% pure [6,7]. The *P. falciparum* NF54, K1, and W2 cell lines were obtained from the Malaria Research and Reference Reagent Resource Center (MR4) at BEI Resources, Manassas, VA, USA. For screening gametocytes, the NF54-PfS16-GFP-Luc and NF54-Mal8p1.16-GFP-Luc lines were kindly provided by Professor David Fidock, Department of Microbiology and Immunology, Columbia University Irving Medical Center, New York, NY 10032. The mammalian liver HepG2 cell line was kindly provided by Professor Duncan Cromarty, Department of Pharmacology, University of Pretoria, SA.

### 4.1. Efficacy

For assays in vitro against blood stage asexual *Pf* parasites, DHA, artesunate, artemether, chloroquine (CQ), and methylene blue (MB) were used as reference drugs. All other assay conditions are as previously described [6,7]. Compound working solutions were prepared from a 10 mM stock solution in 100% (*vol*/*vol*) dimethyl sulfoxide (DMSO, Sigma-Aldrich) in supplemented RPMI 1640 medium containing AlbuMAX II with a final DMSO concentration of 0.1% (*vol*/*vol*), previously determined to be nontoxic to blood stage asexual parasites and gametocytes. The dose-responses of the amino-artemisinins were assayed using a 2-fold serial drug dilution on in vitro 95% ring-stage parasites at 37 °C under 90% N_2_, 5% CO_2_, and 5% O_2_ atmospheric conditions, detecting SYBR green I fluorescence as the proliferative marker following a 96 h drug treatment (1% parasitemia and 1% hematocrit) using a GloMaxR-Multi+ Detection System (Promega) [71,72]. No drug washout steps were performed during drug incubation periods prior to the assays. Activity against the *Pf* drug-sensitive NF54 strain and the drug-resistant K1 (resistant to CQ, quinine, pyrimethamine, and cycloguanil) and W2 (resistant to CQ, quinine, pyrimethamine, and cycloguanil) strains was evaluated. Data analysis was performed using GraphPad Prism (version 6), intra-assay variability was monitored with Z-factors, and acceptable interassay reproducibility was determined from the percent coefficient of variation (CV) [61]. The data for each compound are from at least three independent biological replicates, each performed in technical triplicates, and results are expressed as the compound concentration (IC_50_) at which 50% parasite proliferation is affected.

The same reference compounds with the exception of CQ were used for the early- and late-stage gametocyte assays. Gametocytocidal activity was determined using the transgenic NF54-*pfs*16-GFP-Luc and NF54-Mal8p1.16-GFP-Luc reporter lines [58,59]. Compounds were initially screened for activity at a fixed drug concentration of 1 µM, with dose responses subsequently determined using 2-fold serial drug dilutions for 48 h against early-stage gametocytes (day 5 post-induction population, ≥95% stages II–III) and 10-fold serial drug dilutions for 48 h or 72 h against late-stage gametocytes (day 10 post-induction population, ≥90% late-stage [stages IV–V]) (2 to 3% gametocytemia, 2% hematocrit) at 37 °C under 90% N_2_, 5% CO_2_, and 5% O_2_ [6]. No drug washout steps were performed during the drug incubation periods prior to the assays. In all cases, an interference assay was run in parallel to eliminate the possibility of false positives arising through compound interference with the luciferase readout. The data for each compound are from at least three independent biological replicates, each performed in technical triplicates, and results are expressed as the compound concentration at which 50% parasite viability was affected (IC_50_).

### 4.2. Cytotoxicity

Cytotoxicity against mammalian hepatocellular carcinoma cells (HepG2 cells) was determined using the lactate dehydrogenase assay (LDH, Biovision Inc., Milpitas, CA, USA) [6]. Cells were detached using 0.25% Trypsin-EDTA (Sigma-Aldrich), and viability monitored using Trypan Blue. Prior to assay, cells (10 000/well) were plated in complete DMEM media supplemented with 10% (*vol*/*vol*) heat inactivated fetal bovine serum and incubated overnight at 37 °C at 5% CO_2_ and 95% humidity. Compound working solutions were prepared from 10 mM stock solutions (100% DMSO) in complete DMEM to a final DMSO (Sigma-Aldrich) conc. of <0.1% (*vol*/*vol*). Cells were treated using a 2-fold serial dilution and incubated for 24 h at 37 °C, 5% CO_2_ and 95% humidity. Following incubation, the LDH reagent was added to cells (as per manufacturer’s instructions), incubated at 37 °C at 5% CO_2_ and 95% humidity for 30 min and absorbance read at 450 nm using a SpectraMax Paradigm Multimode Detection Platform (Molecular Devices). Data analysis was performed using GraphPad Prism (version 6) software. The data for each compound are from at least three independent biological replicates, each performed in technical duplicates, and results are expressed as the compound concentration at 50% cell cytotoxicity (IC_50_).

### 4.3. Pharmacokinetics and Metabolism

All work was conducted with prior approval of the animal ethics committee of the University of Cape Town in accordance with the South African National Standards for the Care and Use of Animals for Scientific Purposes and guidelines from the South African Department of Health [73,74]. The requirements and methods for the pharmacokinetics in vivo involving *po* and *iv* drug administration, sample extraction, analyses of samples by LC-MS/MS, and data analyses are described elsewhere [25]. For the animal experiments and formulations, the compounds were dissolved in a 10:60:30 mixture of *N*,*N*-dimethylacetamide-polypropylene glycol-polyethylene glycol for intravenous dosing (*iv*) at 5 mg/kg. For the oral dose (*po*), compounds were individually dosed as a suspension in 0.5% (*w*/*v*) hydroxypropylmethylcellulose in water with 0.2% Tween80 at 50 mg/kg to male C57/BL6 mice (*n* = 3 for each group). For PK sampling, blood samples were collected at predetermined times (0.08, 0.25, 0.5, 1, 3, 5, 7, and 24 h) for both *po* and *iv* dosing via tail bleeding into heparinized tubes, centrifuged, and the plasma samples were stored at −80 °C until extraction. For preparation of samples, the frozen plasma samples were thawed and 15 μL was extracted by liquid–liquid extraction using a universal buffer (pH8) containing 10 ng/mL of the internal standard and ethyl acetate, and the extract was vortexed and centrifuged. Calibration standards and quality controls were extracted following the same procedure. Supernatants were dried down, reconstituted, and injected onto the column for LC-MS/MS analysis using an AB SCIEX 5500 QTRAP instrument coupled to an Agilent 1260 HPLC detection system as previously described [25]. Likewise, data acquisition and evaluation were conducted with Analyst 1.6.2 software (Applied Biosystems, Foster City, CA, USA. For PK analysis, non-compartmental analysis and complementary modelling was performed for the determination of the pharmacokinetic parameters using PK Solutions v2.0 (Summit Research Services) and Kinetica v5.1 (Thermo Fisher Scientific, Tewksbury, MA, USA). Data analysis and presentation was performed using SigmaPlot v14.5 (Systat Software, San Jose, CA, USA).

## 5. Conclusions

The lipophilic artemiside is metabolized to the more polar, water-soluble compounds artemisox, artemisone, and M1. Each compound of the foregoing tetrad is potently active against *Pf* asexual and gametocyte blood stage parasites in vitro, and each displays low levels of toxicity towards HepG2 cells. Within a murine model, administration of artemiside results in substantially higher exposure levels including higher C_max_ values of active drug, when concentrations of the parent artemiside and its metabolites are taken into consideration, compared to administration of artemisone itself. This is handsomely reflected in the approximately 3-fold greater activity of artemiside compared to artemisone against the malaria parasite in vivo and superior activities elicited in the murine CM model. Further, the antimalarial efficacies of artemiside lie well below its toxic threshold, with its neurotoxic potential in particular being substantially less than that of DHA. Next, it will be necessary to fortify the results presented here by conducting comparative ex vivo bioassays of artemiside and artemisone in a primate model according to the method originally used for artemisone [24] in which as noted above activities of artemisone and its metabolites M1 and hydroxylated derivatives were apparent up to 7 h post-administration of a single oral dose of artemisone at 10 mg/kg.

Overall, the data strongly support the proposal that artemiside, which itself is metabolized to artemisone, is eminently suitable for evaluation as an antimalarial drug in a clinical setting. It thereby becomes a prime candidate for substituting current artemisinins in the development of rational new TACTs.

## Figures and Tables

**Figure 1 pharmaceutics-13-02066-f001:**
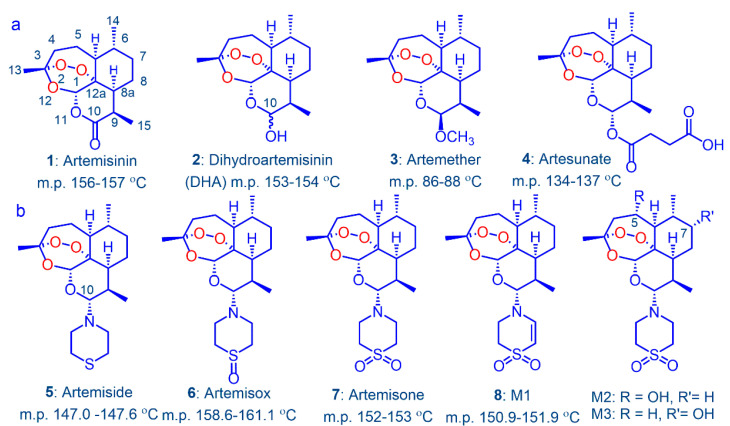
(**a**) Artemisinin **1** and clinical derivatives DHA **2**, artemether **3** and artesunate **4** [9,10,11,12,13]; the last two are rapidly converted into DHA in vivo [14,15,16,17,18,19,20]; (**b**) the highly crystalline thiomorpholine derivative artemiside **5**, the thiomorpholine *S*-oxide artemisox **6**, the thiomorpholine *S*,*S*-dioxide artemisone **7**, and the principal metabolites of artemisone: the unsaturated thiomorpholine-*S*,*S*-dioxide (1,4-dihydrothiazine-*S*,*S*-dioxide) M1 **8**, and the hydroxylated artemisone derivatives M2 and M3 [24].

**Figure 2 pharmaceutics-13-02066-f002:**
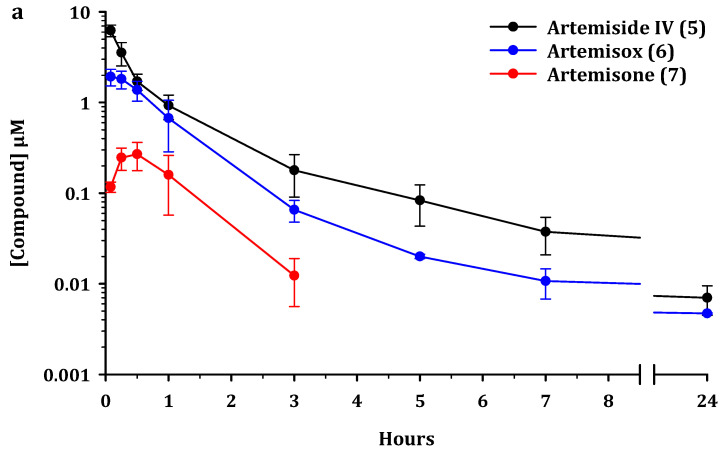
Circulating concentrations of artemiside **5** and its principal metabolites artemisox **6** and artemisone **7** after (**a**). intravenous (*iv*) administration of artemiside at 5 mg/kg and (**b**). oral (*po*) administration at 50 mg/kg to male C57/BL6 mice (*n* = 3 for each group). All results are presented as mean ± standard deviation.

**Figure 3 pharmaceutics-13-02066-f003:**
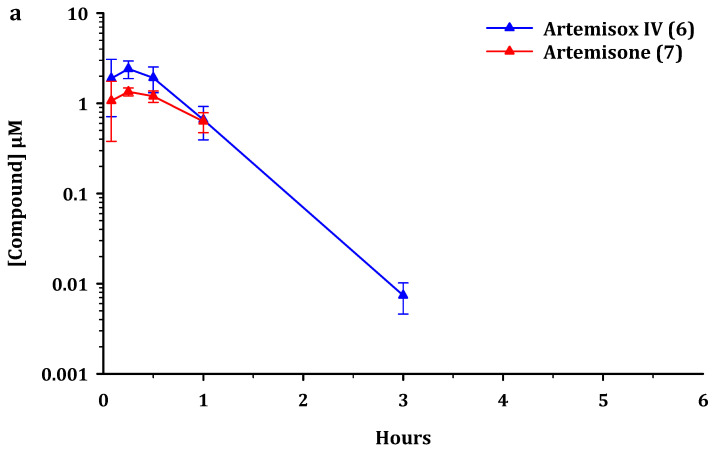
Circulating concentrations of artemisox **6** and its principal metabolite artemisone **7** after (**a**). intravenous (*iv*) administration of artemisox at 5 mg/kg to male C57/BL6 mice (*n* = 3) and (**b**). after oral (*po*) administration at 50 mg/kg to male C57/BL6 mice (*n* = 3). All results are presented as mean ± standard deviation.

**Figure 4 pharmaceutics-13-02066-f004:**
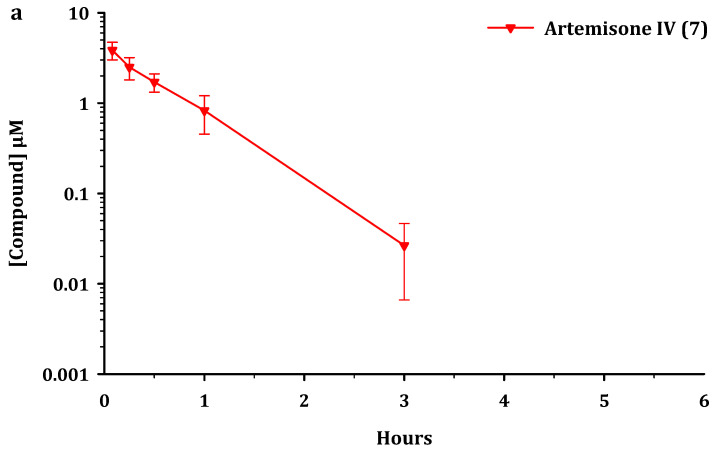
Circulating concentrations of artemisone **7** after (**a**). intravenous (*iv*) administration of artemisone at 5 mg/kg and (**b**). oral (*po*) administration at 50 mg/kg to male C57/BL6 mice (*n* = 3 for each group). All results are presented as mean ± standard deviation.

**Figure 5 pharmaceutics-13-02066-f005:**
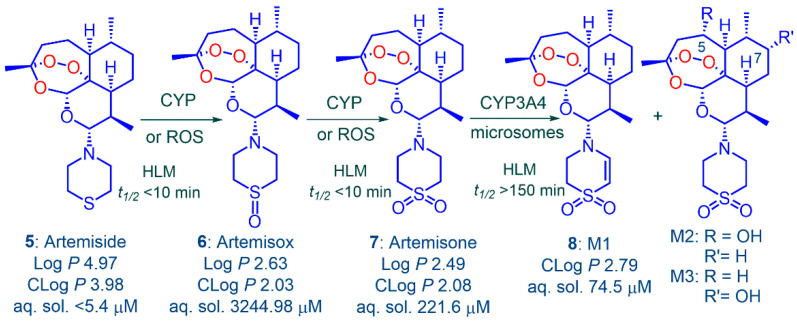
Metabolism of artemiside **5** via enzymatic or non-enzymatic oxidation to artemisox **6** and then to artemisone **7**. Human liver microsomal (HLM) half-lives *t*_1/2_ for each of artemiside, artemisone, and M1 are given [25]. Artemisone is transformed by CYP3A4 into the metabolites M1 **8**, M2, and M3 and smaller amounts of other metabolites [24,25,39]. Measured and calculated Log *P* values (ChemDraw V15.1) and aqueous solubility at pH 7.2 for compounds **5–7** are from [24]; aqueous solubility at pH 7.4 for metabolite M1 **8** is from [25].

**Table 1 pharmaceutics-13-02066-t001:** Activities in vitro against chloroquine-sensitive and multidrug resistant asexual blood stage *Pf* parasites and cytotoxicities against HepG2 cells.

Drug ^a^	IC_50_ nM ^b^					IC_50_ µM ^e^
	NF54	K1	RI ^c^	W2	RI ^d^	HepG2
Chloroquine ^f^	10.0 ± 3.0	154 ± 14	15.4	233 ± 49	23.3	58.4
Methylene Blue ^f^	5.0 ± 0.8	6.45 ± 0.30	1.29	5.13 ± 0.31	1.03	-
DHA **2** ^f^	2.51 ± 0.19	1.51 ± 0.33	0.6	1.74 ± 0.22	0.7	-
Artemether **3** ^f^	1.86 ± 0.17	9 ± 2	4.8	7 ± 1	3.8	-
Artesunate **4** ^f^	3.00± 0.29	4 ± 1	1.3	2.4 ± 0.4	0.8	-
Artemiside **5** ^f^	1.11 ± 0.17	1.6 ± 0.4	1.47	1.75 ± 0.27	1.58	>50
Artemisox **6**	1.95 ± 0.25	1.5 ± 0.5	0.8	1.5 ± 0.4	0.7	>50
Artemisone **7** ^f^	1.2 ± 0.4	1.01 ± 0.19	0.85	1.6 ± 0.4	1.36	>50
M1 **8**	2.63 ± 0.24	1.50 ± 0.23	0.57	2.26 ± 0.08	0.86	>50

^a^ Structures for **2**–**8** in Figure 1; *Pf* NF54 CQ sensitive; K1 CQ, pyrimethamine, mefloquine, cycloguanil resistant; W2 CQ, quinine, pyrimethamine, cycloguanil resistant; ^b^ Data from proliferative SYBR Green I assay with three independent biological replicates, each performed as technical triplicates, ± SEM; ^c^ Resistance index RI = IC_50_ K1/IC_50_ NF54; ^d^ RI = IC_50_ W2/IC_50_ NF54; ^e^ Results for cytotoxicity (LDH assay) against HepG2 cells from three independent biological replicates, performed in technical duplicates, ±SEM; ^f^ Antimalarial efficacy data from refs [6,7].

**Table 2 pharmaceutics-13-02066-t002:** Activities in vitro against early-(EG) and late-(LG) blood stage *Pf* NF54 gametocytes.

Compound ^a^	IC_50_ nM ^b^	
	EG	LG
Methylene Blue ^c^	95.0 ± 11.3	143.0 ± 16.7
DHA **2** ^c^	43.0 ± 3.9	33.66 ± 1.98
Artemether **3** ^c^	37.7 ± 2.0	136.2 ± 85.9
Artesunate **4** ^c^	62.8 ± 3.0	259.4 ± 80
Artemiside **5** ^c^	16.4 ± 1.0	1.5 ± 0.5 ^d^
Artemisox **6**	18.94 ± 0.98	2.72 ± 0.09 ^d^
Artemisone **7** ^c^	1.94 ± 0.11	42.4 ± 3.3 ^d^
M1 8	13.4 ± 2.7	ND ^e^

^a^ Structures of **2**–**8** in Figure 1; ^b^ Luciferase-based assay against Luc reporter cell line with three independent biological replicates, performed as technical triplicates, ±SEM; EG >95% stages II-III; LG >90% stages IV-V, data from 48-h incubation period; ^c^ data from ref. [6]; ^d^ dose responses obtained at 72 h, ^e^ dose response (IC_50_) not determined; M1 showed ≥93% inhibition of LG viability at fixed dose of 1 µM for a 48 h incubation period.

**Table 3 pharmaceutics-13-02066-t003:** Pharmacokinetic parameters calculated from the intravenous administration (*iv*) of 5 mg/kg artemiside **5**, artemisox **6,** and artemisone **7** to male C57/BL6 mice together with their respective metabolites formed in situ ^a^.

Drug *iv*	*t*_½_ h	*MRT_0-last_* h	*CL* L/h/kg	*V_ss_* L/kg	*AUC_0-last_* µmol.h/L	*AUC_0-∞_* µmol.h/L
**5**	1.2 ± 0.1	1.0 ± 0.1	3.5 ± 0.4	4.1 ± 0.5	3.8 ± 0.5	3.9 ± 0.4
**6** from **5**	-	-	-	-	1.9 ± 0.5	-
**7** from **5**	-	-	-	-	0.32 ± 0.12	-
**6**	0.43 ± 0.12	0.57 ± 0.04	6.9 ± 2.0	4.0 ± 1.2	2.0 ± 0.7	2.0 ± 0.7
**7** from **6**	-	-	-	-	1.1 ± 0.2	-
**7**	0.39 ± 0.08	0.56 ± 0.07	5.3 ± 1.5	3.0 ± 0.5	2.4 ± 0.6	2.5 ± 0.6

^a^ *n* = 3; dose of 5 mg/kg equates to 13.5 µmol/kg artemiside **5**, 13 µmol/kg artemisox **6** and 12.4 µmol/kg artemisone **7**, respectively; metabolites **6** and **7** formed from **5**; metabolite **7** formed from **6**; *MRT* mean residence time (0-last, up to 7 h); *CL* clearance; *V_ss_* volume of distribution at steady state; *AUC* area under the concentration-time curve (*0-last*, up to 7 h); mean ± SD.

**Table 4 pharmaceutics-13-02066-t004:** Pharmacokinetic parameters calculated from the oral administration (*po*) of 50 mg/kg artemiside **5**, artemisox **6,** and artemisone **7** to male C57/BL6 mice together with their respective metabolites formed in situ ^a^.

Drug *po*	*t_½_* h	*C_max_* µM	*MRT_0-last_* h	*MAT* h	*AUC_0-last_* µmol.h/L	*F* %	Ratio *AUC_0-last_* Metabolite/Parent
**5**	1.40 ± 0.04	0.13 ± 0.01	1.9 ± 0.1	0.92 ± 0.20	0.36 ± 0.06	1.0 ± 0.3	-
**6** from **5**	-	2.5 ± 0.3	-	-	4.0 ± 0.8	-	11.1 ± 1.5
**7** from **5**	-	1.8 ± 0.1	-	-	2.8 ± 0.3	-	7.8 ± 0.7
**6**	0.54 ± 0.10	4.5 ± 2.3	0.69 ± 0.09	0.13 ± 0.07	3.3 ± 1.4	16 ± 2	-
**7** from **6**	-	3.8 ± 0.6	-	-	4.7 ± 0.6	-	1.5 ± 0.4
**7**	0.39 ± 0.05	1.7 ± 0.4	0.54 ± 0.13	0	1.1 ± 0.1	4.9 ± 1.7	-

^a^ *n* = 3; dose of 50 mg/kg equates to 135 µmol/kg artemiside **5**, 130 µmol/kg artemisox **6** and 124 µmol/kg artemisone **7**, respectively; metabolites **6** and **7** formed from **5**; metabolite **7** formed from **6**; *MRT* mean residence time (0-last, up to 7 h); *MAT* mean absorption time (*MAT* = *MRTpo*–*MRTiv*); *AUC* area under the concentration-time curve (0-last, up to 7 h); *F* bioavailability; mean ± SD.

## Data Availability

All data is published here and in the Appendix A.

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
