# Peer review of "The Artemiside-Artemisox-Artemisone-M1 Tetrad: Efficacies against Blood Stage P. falciparum Parasites, DMPK Properties, and the Case for Artemiside"

_pharmaceutics, 2021, doi:10.3390/pharmaceutics13122066_

Round 1
Reviewer 1 Report
The authors of the manuscript describe a potential antiplasmodial compound - the endoperoxide artemisid that undergoes metabolism first to artemisox followed by artemisone and a further metabolite M1. In contrast to other/currently used artemisinin derivatives it is not metabolized to dihydroartemisinin. The in vitro activities against asexual Plasmodium falciparum blood stages and early and late gametocyte stages of the four different compounds were evaluated and presented, in addition pharmacokinetic properties of the first three compounds were evaluated in a mouse model.
Overall the manuscript is clearly written and well-structured, including the relevant references. The discussion mentions rather extensively previous findings, including activities against other pathogens (Toxoplasma).
Some minor comments:
1)Table 1: According to this table the cytotoxicity of the compounds is very high. Are these values in nM, as indicated or is this a typo?
2) Table 2: The compounds are quite active against late stage gametocytes. This is known to be the case for the used assay. However, it should be stated in the discussion that in other in vitro late stage gametocyte assays (e.g. ATP based) artemisinins/endoperoxides show nearly no activity. And also in vivo the activity of artemisinin derivatives against late stage gametocytes is not proven.
From my point of view, it has to be verified in further assays that these compounds have a pronounced late stage gametocyte activity. Please add a paragraph to the discussion on this.
Figure 2: Can the authors show in the supplement a figure for artesunate and dihydroartemisinin in comparison? This is not absolutely necessary, but if available would help to compare pharmacokinetics of these new compounds to artesunate/DHA in a mouse model.
If a comparable figure is not available, I would suggest to mention it in the discussion for comparison.
Table 3: Please explain once in the table legend what is meant by “6 from 5”
Author Response
Response to Reviewer 1
Comments and Suggestions for Authors
The authors of the manuscript describe a potential antiplasmodial compound - the endoperoxide artemisid that undergoes metabolism first to artemisox followed by artemisone and a further metabolite M1. In contrast to other/currently used artemisinin derivatives it is not metabolized to dihydroartemisinin. The in vitro activities against asexual Plasmodium falciparum blood stages and early and late gametocyte stages of the four different compounds were evaluated and presented, in addition pharmacokinetic properties of the first three compounds were evaluated in a mouse model.
Overall the manuscript is clearly written and well-structured, including the relevant references. The discussion mentions rather extensively previous findings, including activities against other pathogens (Toxoplasma).
Some minor comments and authors' responses:
We thank the reviewer for picking up an important correction, and for the time and effort spent in providing suggestions for improving the MS. We have addressed all the concerns except as noted below.
1)Table 1: According to this table the cytotoxicity of the compounds is very high. Are these values in nM, as indicated or is this a typo?
Response: We do thank the reviewer for pointing this out. This was a typing error. The concentration range for the HepG2 IC50 values were determined in the µM range, which has been corrected in Table 1. However, in order to correlate the data with previous published reports (Lelièvre, J.; Almela, M.J.; Lozano, S.; Miguel, C.; Franco, V.; Leroy, D.; Herreros, E. Activity of clinically relevant antimalarial drugs on Plasmodium falciparum mature gametocytes in an ATP Bioluminescence “Transmission Blocking” Assay. PLoS ONE 2012, 7, e35019, doi: 10.1371/journal.pone.0035019), we omitted the µM concentrations indicated and have referred to them as >50 µM. The selectivity index has also been removed from table 1 as the high selectivity is clear based on the reported antimalarial IC50 values. Thus, the units in the column heading for HepG2 have been changed to µM, and the column with selectivity indices has been deleted. We have inserted the comments at lines 133-134 in the new MS: "In line with previous reports involving assessment of cytotoxicities of antimalarial drugs against the mammalian liver HepG2 cell line [44], activities of the amino-artemisinins in Table 1 are some five orders of magnitude less than against the malarial parasites." The above PLoS reference is now inserted as reference 44 in the new MS.
2) Table 2: The compounds are quite active against late stage gametocytes. This is known to be the case for the used assay. However, it should be stated in the discussion that in other in vitro late stage gametocyte assays (e.g. ATP based) artemisinins/endoperoxides show nearly no activity. And also in vivo the activity of artemisinin derivatives against late stage gametocytes is not proven.
From my point of view, it has to be verified in further assays that these compounds have a pronounced late stage gametocyte activity. Please add a paragraph to the discussion on this.
Response: The reviewer has appropriately highlighted a complicated issue. Although we report nM activities of artemisone, artemiside and artemisox against late-stage gametocytes we have to stress that these IC50 values could only be obtained at 72 h incubation periods (Coertzen, Wong et al., refs 6 and 7) and not at 48 h as is observed with other artemisinin derivatives (DHA, artemisinin, artesunate and artemether) in Table 2 (also referenced from Coertzen and Wong et al., refs 6 and 7). This indicates that artemisone, artemiside and artemisox display activity only at extended incubation periods for late-stage gametocytes (mixture of stage IV and V gametocytes). Similar activities for artemisone on late-and mature-stage gametocytes using a luciferase based assay have also been reported for 72 h incubation periods, as we highlight in the discussion in Section 3.3 (new refs 59 and 60) and correlate with what we observed for artemisone and artemiside in ref 6. Artemisone has also been reported to show potent activities on the pLDH assay platform based on a 72 h drug incubation, but poor activity at a 48 h using the Acridine orange-GMT assay (ref. 61). In summary, our data correlates to previous reports where clinical artemisinins show activity on gametocyte populations with stage IV gametocytes at 48 h incubation, but requires a 72-96 h incubation against mature stage V gametocytes for activity. Thus, we have slightly amended the Abstract to highlight the incubation period dependency for late-stage gametocytes (lines 29-30), and have discussed this in more detail in the amended MS (Section 3.3, lines 336-351).
Figure 2: Can the authors show in the supplement a figure for artesunate and dihydroartemisinin in comparison? This is not absolutely necessary, but if available would help to compare pharmacokinetics of these new compounds to artesunate/DHA in a mouse model.
If a comparable figure is not available, I would suggest to mention it in the discussion for comparison.
Response: We thank the reviewer for this comment, for indeed it is pertinent to be able to draw a direct comparison with one or more of the current clinical artemisinin derivatives with artemiside or artemisone. The only directly comparable data available is that for artemether, which is presented in reference 25 in the amended MS. Thus, we have inserted the following comments into the MS, and have added four new references 52-55 in the new MS lines 308-318 which we do hope expresses the idea: "In the case of artemether, we previously presented data indicating its rapid metabolism to DHA using oral administration at the same dose with the same murine model as described here (see Table 3, ref [25]). Summation of Cmax values for artemiside and its metabolites artemisone and M1 is approximately 1.67 times greater than that for artemether (Cmax 0.6 µM, ref [25]). However, no measurement of Cmax for artemisox formed in situ was carried out in that study [25]. From the current study, summation of Cmax values for artemiside and its metabolites artemisox and artemisone at 4.43 µM (Table 4) is some 7.4 times greater than Cmax for artemether obtained from the earlier study. Thus overall, the substantially higher exposure of artemiside and its active metabolites coupled also with their potency (Table 1) should result in enhanced activity with respect to artemether itself. We do not have comparable data for artesunate, but as noted above for artemisone itself, ex vivo bioassay in a primate model indicates that activity is observable up to 7 hours after administration of a single dose of artemisone (10 mg/kg). Equivalent activity of a single dose of artesunate (10 mg/kg) is observable up to 1 hour post dose where the active metabolite must be DHA [24]. Thus it is clear that use of artemiside in this model will provide better results."
Table 3: Please explain once in the table legend what is meant by “6 from 5”
Response: We do regret this is not clear. As we comment upon in the text, and as is apparent from the figures 2-4 and figure legends, we use this to indicate that we have measured plasma concentrations of the metabolite 6 (artemisox) formed from artemiside 5 (the numbers designate the compounds in Figure 1). Nevertheless, we have added "together with their respective metabolites formed in situ." to the Tables 3 and 4 headings, and have inserted "metabolites 6 and 7 formed from 5; metabolite 7 formed from 6;" in the footnotes to these Tables. We do hope this will be acceptable.
Submission Date 01 November 2021
Date of this review 11 Nov 2021 14:48:46

Reviewer 2 Report
The authors evaluated the in vitro antimalarial activity, cytotoxicity, and PK characteristics of thiomorpholine artemisinin derivatives (artemiside, artemisone, artemisox) that are not metabolized to dihydroartemisinin (like all currently available derivatives, including artemether and artesunate), with the aim to identify the best candidate for triple ACT.
The methods are described clearly and in detail. The results include many useful supplementary materials and are well presented. Artemisox was highly active in vitro against three Pf reference strains, with IC50 values in nanomolar range similar to those of other artemisinin derivatives. The metabolite M1 was also active in vitro. The PK results are presented together with those of other similar derivatives, rendering comparison of data easier to understand. The results tend to support the authors’ argument that artemiside-artemisone may be a promising candidate for further development. The paper is very well written.
Major comments:
None
Minor comments:
Line 47, 48 and elsewhere: artemisinin-based combination therapies (ACTs)
Line 94: liquid chromatography-tandem mass spectrometry (LC-MS/MS)
Line 97: P. falciparum (Pf) in vitro
Table 1: The table title only refers to Pf and does not mention cytotoxicity data. Please revise the title and include cytotoxicity.
Line 120, Table 1 legend: footnote “(f)” IC50 W2/IC50 NF54 should be “(d)”. There is another footnote (f) for SI, which is correct.
Line 154: Structures (space) of
Table 2: To be consistent, all IC50 values can be expressed to the first decimal place: DHA, 33.7; artemisox, 18.9 ± 1.0 ; 2.7 ± 0.1; artemisone, 1.9 ± 0.1. It may be preferable to state “ND” for “not determined” in the table for M1 vs LG, instead of “93% @ 1 µM;” the table legend for the footnote (e) is explicit enough.
Line 162: delete one of two “to”
Figure 5: We usually do not expect to see a figure in the Discussion section. The authors may want to consider placing this figure as a supplementary material.
Line 315, references: In line 291, reference [50] was cited. In line 315, should the reference number be [51] instead of [65]? Please check the reference numbers.
Line 320, “liver stage P. berghei sporozoites”: Do the authors mean liver stage P. berghei parasites? See also Table S4c table title and table legend.
Line 342: recrudescence with eventual mortality was observed
Lines 347-348, “As for Pf, the compounds…display efficacies at least an order of magnitude inferior to those for Pf”: Please re-check the meaning of this sentence. Is it “inferior to those for Pf or for T. gondii? The EC50 in vitro for toxo ranged from 0.11 to 0.21 µM (reference: Dunay et al. 2009).
Line 352: “per day” is redundant. It can be deleted. The daily dose is given in the same line (10 mg/kg/d).
Lines 354-371: This long paragraph on Toxoplasma gondii can be shortened, especially the findings reported using interferon-gamma negative mice in Ref 57.
Line 374: “each of” can be deleted.
Line 508: data strongly support
Supplementary data Table S4a, S4b, S5: In the present work, SYBR Green I-based in vitro assay was used. Data presented in S4 were based on tritiated hypoxanthine-based assay. The reference(s) for the latter technique should probably be cited in the list of reference in supplementary data. Peter’s four-day test is also described in supplementary data. A reference citation would be useful.
Ref 1: Please cite the latest available WHO World Malaria Report (2020).
References: The format should be re-checked, including journal abbreviations: Ref 2 Lancet Infect Dis; Ref 8 Commun Chem; Ref 18, 22, 54, 55, 64 Malar J; Ref 19 Int J Infect Dis; Ref 24, 34 Angew Chem Int Ed Engl; Ref 28 Neurotox Res; Ref 46 Free Radic Biol Med; Ref 47 BMC Biochem; Ref 48 Redox Biol; Ref 61 Trans R Soc Trop Med Hyg.
Ref 4 is incomplete. Curr Epidemiol Rep. 2021:1-17. Moreover, it is an Epub ahead of print. Please update it, if possible.
Ref 27 seems to be incomplete. Sci Signal 2015, 8, ec118
Ref 31 is published. Please update it: Int J Parasitol Drugs Drug Resist 2021, 17, 186-190.
Ref 32: If the authors follow their format, it should be “et al.” after Martin, N.J. Also “Plasmodium falciparum” in the article title.
Ref 37 is a book chapter. Please provide the complete reference.
Ref 40: pheroid
Ref 66: Please add its web link.
Ref 67: The web link can be added to facilitate access to this document by interested readers: https://ahrecs.com/resources/ethics-health-research-principles-processes-structures-2nd-ed-south-africa/
Author Response
Response to Reviewer 2
Comments and Suggestions for Authors
The authors evaluated the in vitro antimalarial activity, cytotoxicity, and PK characteristics of thiomorpholine artemisinin derivatives (artemiside, artemisone, artemisox) that are not metabolized to dihydroartemisinin (like all currently available derivatives, including artemether and artesunate), with the aim to identify the best candidate for triple ACT.
The methods are described clearly and in detail. The results include many useful supplementary materials and are well presented. Artemisox was highly active in vitro against three Pf reference strains, with IC50 values in nanomolar range similar to those of other artemisinin derivatives. The metabolite M1 was also active in vitro. The PK results are presented together with those of other similar derivatives, rendering comparison of data easier to understand. The results tend to support the authors’ argument that artemiside-artemisone may be a promising candidate for further development. The paper is very well written.
Major comments:
None
Minor comments:
We thank the reviewer for the efforts in improving the MS. We have addressed all the concerns except as noted below.
Line 47, 48 and elsewhere: artemisinin-based combination therapies (ACTs)
Response: the expressions have been amended
Line 94: liquid chromatography-tandem mass spectrometry (LC-MS/MS)
Response: this has been corrected.
Line 97: P. falciparum (Pf) in vitro
Reponse: this has been corrected.
Table 1: The table title only refers to Pf and does not mention cytotoxicity data. Please revise the title and include cytotoxicity.
Response: We thank the reviewer for noting this; we have amended the title of Table 1 to: Activities in vitro against chloroquine-sensitive and multidrug resistant asexual blood stage Pf parasites and cytotoxicities against HepG2 cells.
Line 120, Table 1 legend: footnote “(f)” IC50 W2/IC50 NF54 should be “(d)”. There is another footnote (f) for SI, which is correct.
Response: We thank the reviewer for noting the problem, and have corrected this.
Line 154: Structures (space) of
Response: noted and corrected
Table 2: To be consistent, all IC50 values can be expressed to the first decimal place: DHA, 33.7; artemisox, 18.9 ± 1.0 ; 2.7 ± 0.1; artemisone, 1.9 ± 0.1. It may be preferable to state “ND” for “not determined” in the table for M1 vs LG, instead of “93% @ 1 µM;” the table legend for the footnote (e) is explicit enough.
Response: We are indeed cognisant of the scientific guidelines relating to the 'number of significant figures', and all data are entered in accordance with these guidelines; we cannot change the data as presented here, in our earlier publications (refs. 6-8), or in numerous other publications related to our own work by other groups. For the entry for M1, we agree with the reviewer and have amended Table 2 accordingly.
Line 162: delete one of two “to”
Response: noted
Figure 5: We usually do not expect to see a figure in the Discussion section. The authors may want to consider placing this figure as a supplementary material.
Response: Well, we regard this as normal practice, and we cannot remove the figure based on the accompanying discussion.
Line 315, references: In line 291, reference [50] was cited. In line 315, should the reference number be [51] instead of [65]? Please check the reference numbers.
Response: We thank the reviewer for noting this. This has been corrected by placing in the correct location, and become now reference 58 in the amended MS.
Line 320, “liver stage P. berghei sporozoites”: Do the authors mean liver stage P. berghei parasites? See also Table S4c table title and table legend.
Response: 'Sporozoites' represent liver stage parasites of P. berghei, but in any event, we have changed the description to "the liver sporozoite stage of P. berghei parasites" (lines 349-350 in the amended MS, and in Table S4c).
Line 342: recrudescence with eventual mortality was observed
Response: noted and corrected
Lines 347-348, “As for Pf, the compounds…display efficacies at least an order of magnitude inferior to those for Pf”: Please re-check the meaning of this sentence. Is it “inferior to those for Pf or for T. gondii? The EC50 in vitro for toxo ranged from 0.11 to 0.21 µM (reference: Dunay et al. 2009).
Response: we thank the reviewer for noting this ambiguity. We have changed the expression to "As for Pf, the compounds are equipotent in vitro, although activities are at least an order of magnitude inferior to activities against Pf. Thus, EC50 values of 0.108 µM and 0.120 µM respectively are recorded for each of artemiside and artemisone against Tg in vitro (cf. artesunate 0.213 µM)."
Line 352: “per day” is redundant. It can be deleted. The daily dose is given in the same line (10 mg/kg/d).
Response: noted and corrected.
Lines 354-371: This long paragraph on Toxoplasma gondii can be shortened, especially the findings reported using interferon-gamma negative mice in Ref 57.
Response: It is important to retain descriptions of the in vivo activities, and we do much prefer to leave as is, especially in relation to the preceding description involving the malaria parasite, and conclusion to the Tg paragraph.
Line 374: “each of” can be deleted.
Response: noted and corrected.
Line 508: data strongly support
Response: noted and corrected
Supplementary data Table S4a, S4b, S5: In the present work, SYBR Green I-based in vitro assay was used. Data presented in S4 were based on tritiated hypoxanthine-based assay. The reference(s) for the latter technique should probably be cited in the list of reference in supplementary data. Peter’s four-day test is also described in supplementary data. A reference citation would be useful.
Response: We do thank the reviewer for pointing this out. However, we would like to avoid too much clutter in the Supplementary material, and to retain consistency in the presentation – thus, we do note that the Desjardins et al. method (Desjardins, R.E.; Canfield, C.J.; Haynes, J.D.; Chulay, J.D.. Quantitative assessment of antimalarial activity in vitro by a semiautomated microdilution technique. Antimicrob. Agents Chemother. 1979, 16, 710–718, doi: 10.1128/AAC.16.6.710) for obtaining IC50 data using the tritiated hypoxanthine assay is used, but this is recorded in the references S2 and S3. Likewise, the Peters 4 day test (ref 36 in ref S4: W. Peters, W.; Robinson, B.L. in Handbook of Animal Models of
Infection (Eds.: O. Zak, M. Sande), Academic Press, London, 1999, Section VI, Parasitic Infection Models, chap. 92, pp. 756-771) is cited in reference S4. The implication of then having to add point references for PK and other methods presented in the Supplementary Material is not appropriate.
All points for the references as noted below have been attended to, with the exception of Ref. 37 – we do thank the reviewer for bringing this to our attention - this is not a book chapter, but it is in a journal which we discover now is no longer available; the Bayer authors considered at the time this journal to be appropriate for the neurotoxicity data. Very unfortunately, there is no doi number, and it does appear that the file is not accessible – we been unable to locate the text through Google Scholar for example. Thus, to provide access to the reference, we have prepared this as Supplementary Material 2, have added a referral to the reference 37 in the amended MS, and have deposited the Supplementary Material 2 together with Supplementary Material 1 – the latter being the original deposition. We do hope this is acceptable.
For ref 40, 'Pheroid' is a proprietary name, so must be left as is.
Ref 1: Please cite the latest available WHO World Malaria Report (2020).
References: The format should be re-checked, including journal abbreviations: Ref 2 Lancet Infect Dis; Ref 8 Commun Chem; Ref 18, 22, 54, 55, 64 Malar J; Ref 19 Int J Infect Dis; Ref 24, 34 Angew Chem Int Ed Engl; Ref 28 Neurotox Res; Ref 46 Free Radic Biol Med; Ref 47 BMC Biochem; Ref 48 Redox Biol; Ref 61 Trans R Soc Trop Med Hyg.
Ref 4 is incomplete. Curr Epidemiol Rep. 2021:1-17. Moreover, it is an Epub ahead of print. Please update it, if possible.
Ref 27 seems to be incomplete. Sci Signal 2015, 8, ec118
Ref 31 is published. Please update it: Int J Parasitol Drugs Drug Resist 2021, 17, 186-190.
Ref 32: If the authors follow their format, it should be “et al.” after Martin, N.J. Also “Plasmodium falciparum” in the article title.
Ref 37 is a book chapter. Please provide the complete reference.
Ref 40: pheroid
Ref 66: Please add its web link.
Ref 67: The web link can be added to facilitate access to this document by interested readers: https://ahrecs.com/resources/ethics-health-research-principles-processes-structures-2nd-ed-south-africa/
Submission Date 01 November 2021
Date of this review 08 Nov 2021 13:34:20

Reviewer 3 Report
Dear authors,
thank you for the very interesting and important work! The new findings can be a milestone in an adapted malaria therapy.
There were no flaws detected in this manuscript. Neither in the approach, nor in the methodology, nor in writing or illustration.
The manuscript should be ready for publishing.
Author Response
We thank the reviewer for examining the manuscript, and the support the reviewer provided. There are no responses required.